# Follicular Atresia, Cell Proliferation, and Anti-Mullerian Hormone in Two Neotropical Primates (*Aotus nancymae* and *Sapajus macrocephalus*)

**DOI:** 10.3390/ani13061051

**Published:** 2023-03-14

**Authors:** Deise de Lima Cardoso, Diva Anélie de Araújo Guimarães, Pedro Mayor, Maria Auxiliadora Pantoja Ferreira, Leandro Nassar Coutinho, Frederico Ozanan Barros Monteiro

**Affiliations:** 1Programa de pós-graduação em Ciência Animal, Universidade Federal do Pará, Rua Augusto Corrêa 01, Belém 66075-110, Brazil; 2Programa de Pós-Graduação em Saúde e Produção Animal na Amazônia, Universidade Federal Rural da Amazônia, Av. Perimetral 2501, Belém 66077-830, Brazil; 3Departament de Sanitat i Anatomia Animals, Facultat de Veterinària, Campus de la UAB, Universidade Autónoma de Barcelona, Plaça Cívica, 08193 Barcelona, Spain

**Keywords:** owl monkey, night monkey, capuchin monkey, ovary, folliculogenesis, apoptosis

## Abstract

**Simple Summary:**

Reproductive processes in Neotropical primates are still poorly understood. Complications arising in these processes may result in a compromised reproductive condition for a significant portion of these species. We evaluated the follicular atresia, cell proliferation, and anti-Mullerian hormone presence in two Neotropical primates, and in three sexual phases (follicular, luteal, and gestational). Our results contribute to a greater understanding of the ovary morphofunctional processes that act in follicular selection, recruitment, and degeneration in these species.

**Abstract:**

This study evaluated the follicular atresia, cell proliferation, and anti-Mullerian hormone action in *Aotus nancymae* and *Sapajus macrocephalus* during three sexual phases (follicular, luteal, and gestational). Follicular quantification and immunolocalization of Caspase-3 protein, B-cell lymphoma 2 (BCL-2), proliferating cell nuclear antigen (PCNA), and anti-Mullerian hormone (AMH) were performed. A significant difference in the quantification between preantral and antral follicles, with a progressive decrease in the antrals, was identified. Protein and hormonal markers varied significantly between follicle cell types (*A. nancymae p* = 0.001; *S. macrocephalus*, *p* = 0.002). Immunostaining in the preantral and antral follicles was present in all sexual phases; for Caspase-3, in granulosa cells, oocytes, and stroma; for BCL-2, in granulosa cells, oocytes, and theca; and for PCNA and AMH, in oocytes and granulosa cells. The immunostaining for Caspase-3 was more expressive in the preantral follicles (follicular phase, *p* < 0.05), while that for BCL-2 and PCNA was more expressive in the antral follicles of the follicular phase. The AMH was more expressive in the primary and antral follicles of nonpregnant females, in both the follicular and luteal phases. Our results contribute to understanding the ovarian follicular selection, recruitment, and degeneration of these species.

## 1. Introduction

In recent years, there has been an increase in reproductive studies focusing on owl or night monkeys (*Aotus* spp.) [1,2,3,4,5,6] and capuchin monkeys (*Sapajus* spp. and *Cebus* spp.) [4,7,8]. There are behavioral differences between the primates, such as their mating systems. Owl monkeys are monogamous [9], while capuchin monkeys are polygamous [10]. Their shared reproductive characteristics include the presence of a menstrual cycle [4] and the ability to easily reproduce in captivity [11]. In both species, it has been observed that during follicular maturation, granulosa cells, theca cells, and oocytes develop a process of cellular death, resulting in follicular atresia and selection, which regulates the ovulation rate and restricts their reproductive potential [12,13].

Apoptosis can be initiated by activation signals such as cleaved Caspase-3, which is an effector protease responsible for cleaving structural and functional proteins in cells [14,15,16]. These mechanisms can be blocked by regulators such as BCL-2 proteins, which prevent the activation of caspases [17,18]. Proliferating cell nuclear antigen (PCNA) is important for DNA synthesis and repair during the cell cycle, therefore maintaining tissue integrity [19,20]. Thus, PCNA can be used to identify cell proliferation in the ovaries of nonhuman primates (NHP) [21].

Anti-Mullerian hormone (AMH) is a growth factor produced by granulosa cells that regulate folliculogenesis [22,23] and plays an inhibitory role in maintaining the primordial ovarian follicle pool [22]. AMH may serve as an indicator of the antral follicle reserve in humans and NHP [23,24].

The processes of follicular activation and selection in *Aotus* spp. and *Sapajus* spp. are still poorly understood [11]. Complications arising in these processes may result in a compromised reproductive condition for a significant portion of these species. Therefore, this study aims to describe the physiology related to follicular atresia through the immunostaining of Caspase-3, BCL-2, PCNA, and AMH proteins in the Ma’s night monkey and the large-headed capuchin.

## 2. Materials and Methods

### 2.1. Study Sites and Animals

Genital organs from postpubertal Ma’s night monkeys (*Aotus nancymae*, *n* = 6) and large-headed capuchins (*Sapajus macrocephalus*, *n* = 6) were collected from two sites in the Peruvian Amazon. Ovaries from six Ma’s night monkeys kept in captivity at the Primate Center of the Instituto Veterinario de Investigaciones Tropicales y de Altura (IVITA, Iquitos, Peru) were collected. An additional six ovaries from large-headed capuchins were collected as waste organs from subsistence hunters in the Yavarí-Mirín River basin (YMR, S 04° 19.53; W 71° 57.33). All biological samples were free of charge and donated voluntarily.

### 2.2. Histological Procedure and Follicular Quantification

Genital organs were examined for evidence of embryos or fetuses. Females with an embryo or fetus were considered pregnant. Nonpregnant females with ovaries containing active cyclic corpus luteum (CL) were classified as being in the luteal phase. Females with ovaries with large antral follicles and without cyclic CL were in the follicular phase of the menstrual cycle.

The ovaries were fixed in 4% buffered formalin (*v*/*v*), dehydrated, embedded in paraffin wax, serially sectioned at 5 μm along the longitudinal axis, stained in hematoxylin and eosin, and analyzed under a photomicroscope. One section was selected for each series of ten sections cut.

Ovarian follicles were classified following the criteria shown in Table 1. The total number of follicles was calculated using the following formula [25]:TN = number of follicles × total number of cuts per ovary × slice thicknessnumber of sections analyzed × nuclear diameter of each follicle(1)

### 2.3. Immunohistochemistry for Caspase-3, BCL-2, PCNA, and AMH

The sections of the ovaries were deparaffinized in xylene and rehydrated in decreasing concentrations of ethanol. The sections were washed in tween saline buffer solution (PBSw) and immersed in 3% hydrogen peroxide in methanol for 30 min. The slides were then incubated in sodium citrate buffer (0.1M pH6.0), heated to 70 °C for 25 min, and incubated in 10% blocking buffer (10% goat serum + 3% bovine serum-BSA albumin + PBSw) for 1 h. Each slide was incubated with primary antibodies in the following dilutions: anti-cleaved Caspase-3 (1:200), anti-BCL-2 (1:200), anti-PCNA (1:100), and anti-AMH (1:100), for 12 h. Subsequently, the samples were post-incubated in peroxidase-conjugated anti-rabbit IgG secondary antibody (1:500) and developed in DAB solution (3,3′diaminobenzidine) for 5 min for photomicroscopic analysis. For the negative control, histological sections of the ovaries in the same analyzed phases were used, with the primary antibody replacement by PBSw.

### 2.4. Hormonal, Proliferative and Cell Apoptotic Quantification

Immunostained oocytes, granulosa cells, and ovarian follicles (preantral and antral) were counted. A total of one thousand elements were counted per ovary. In each ovary, we calculated the percentage of the elements (ovarian follicles, granulosa cells, and oocytes) immunostained for cleaved Caspase-3, BCL-2, PCNA, and AMH [26,27].

number of immunostained follicles/total follicles counted × 100;number of immunostained granulosa cells/total of 1000 granulosa cells analyzed;number of immunostained oocytes/number of oocytes counted × 100.

All analyses were performed using NIS-Elements Microscope Imaging software (Nikon Corporation, Tokyo, Japan) and the Nikon Eclipse Ci-S H550S photomicroscope coupled to a digital camera (Nikon S-Ri1 DS-U3, Tokyo, Japan).

### 2.5. Statistical Analysis

We compared the percentage of immunostained elements (ovarian follicles, granulosa cells, and oocytes) for Caspase-3, BCL-2, PCNA, and AMH counted in each ovary according to the sexual phase of females (follicular, luteal, and pregnant). Differences observed among these phases were assessed by a Kruskal–Wallis test, considering a significance of *p* < 0.05. Data were presented as mean and standard deviation.

The number of granulosa cells, follicles, and oocytes for Caspase-3, BCL-2, PCNA, and AMH were evaluated by the Kruskal–Wallis test, indicating a significant difference between the values quantified by phases. To find the number of viable follicles by follicular types (preantral and antral), a Dunn test was applied, which demonstrated which reproductive periods were different from each other, following the assumptions of normality and homoscedasticity.

A principal coordinate analysis (PCoA) was used to evaluate the relationship between the indicators of granulosa cells, follicles, and oocytes with positive staining for Caspase-3, BCL-2, PCNA, and AMH, while considering the sexual phases to elucidate the action of these proteins in the ovaries [28]. A permutational multivariate analysis of variance (PERMANOVA) was applied to assess the significance of the variation observed in this quantification [29]. The homogeneity of the dispersion and the difference between groups was determined by permutational analysis of multivariate dispersion (PERMDISP) [30]. All statistical analyses were performed using BioEstat 5.0 software (IDSM, MCT, CNPq, Belém, Pará, Brazil) and R Software (R Development Core Team, 2016).

## 3. Results

### 3.1. Follicular Quantification

The number of follicles per follicular type did not vary between sexual phases (*p* > 0.05). Additionally, we observed a significant reduction in the number of follicles as follicular development progressed from primordial follicles to large antral follicles (Table 2).

### 3.2. Immunohistochemistry for Cleaved Caspase-3, BCL-2, PCNA, and AMH

The immunoreactive pattern observed for cleaved Caspase-3, BCL-2, PCNA, and AMH was similar in the two species studied. Immunostaining for cleaved Caspase-3 was observed in oocytes and some granulosa cells of the preantral follicles (Figure 1A); it was also observed in the oocytes, granulosa cells, and in the internal and external theca of the antral follicles (Figure 1E). Strong immunostaining for BCL-2 and PCNA was observed in the oocytes and granulosa cells of the preantral and antral follicles (BCL-2: Figure 1B,F; PCNA: Figure 1C,G). Immunostaining for AMH was observed in granulosa cells of the preantral follicles (Figure 1D), and in oocytes and granulosa cells of the antral follicles (Figure 1H).

### 3.3. Quantification of Immunolabeled Cells for Caspase-3-Cleaved, BCL-2, PCNA, and AMH

We observed significant differences in the quantification of immunostained cells in the ovarian follicles of both species according to the sexual phase (*p* < 0.05). Immunostaining for Caspase-3 and AMH in oocytes, granulosa cells, preantral and antral follicles was higher in females in the follicular phase. Immunostaining for BCL-2 and PCNA was mostly higher in females in the follicular phase (*p* < 0.05); Table 3 shows these differences according to cell type and the NPH.

In both species, the quantification analysis for PCNA showed a significant difference between granulosa cells in the follicular and gestational phases. Higher immunostaining (*p* < 0.05) was observed for the follicular phase and between oocytes and follicles in the follicular and luteal phases, with greater immunostaining for the gestational phase. In contrast, immunostaining for AMH revealed a statistically significant difference in granulosa cells, follicles, and oocytes between the follicular and luteal phases, with higher immunostaining for the follicular phase than the luteal phase (Table 3).

In both the species studied, the percentage of immunostaining for Caspase-3 was higher in the preantral follicles than in the antral follicles (Figure 2). In contrast, the percentage of immunostaining for BCL-2, PCNA, and AMH was higher in the antral follicles than in the preantral follicles (*p* < 0.01) (Figure 2).

Significant differences were observed in the immunostaining patterns of granulosa cells, follicles, and oocytes (Ma’s night monkeys: Pseudo F = 14.486 *p* = 0.001; large-headed capuchins: Pseudo F = 8.32166; *p* = 0.002). In both species, granulosa cells were the most immunoreactive for cleaved Caspase-3, BCL-2, PCNA, and AMH (Table 3 and Figure 3). There were greater differences observed between the quantification of granulosa cells in relation to follicles and oocytes (Pseudo F = 14.486; *p* = 0.001; Figure 3A,B).

The quantification of immunohistochemical markers explained 94.52% of the positive immunostaining of follicles, granulosa cells, and oocytes for cleaved Caspase-3, BCL-2, PCNA, and AMH in Ma’s night monkeys (Figure 4A) and 94.85% for the same in large-headed capuchins (Figure 4B). In Ma’s night monkeys, the variables that contributed most to the formation of the first axis (75.83%) were BCL-2, PCNA, and AMH, which were positively related to the axis. For the second axis, cleaved Caspase-3 contributed positively with 18.69%. In large-headed capuchins, the variables that contributed most to the formation of the first axis (71.03%) were Caspase-3 and BCL-2. For the second axis (23.82%), PCNA and AMH contributed positively. The results presented in Figure 3 confirmed this ordering (Ma’s night monkeys: Pseudo F = 14.486; *p* = 0.001) and (large-headed capuchins: Pseudo F = 8.32166; *p* = 0.002).

## 4. Discussion

A greater number of preantral follicles (primordial and primary) was observed in relation to antral follicles in the two species studied. These data are consistent with that previously observed in women [25,31] and in capuchin monkeys [32]. Preantral follicles, in the early development stages, form the reserve pool. Such structures undergo selection and dominance processes, in waves of follicular degeneration, which determine the reproductive potential of females [31,33]. Therefore, we believe that the quantity of preantral follicles present is directly related to ovarian reserve maintenance in both species in this study. This has been reported previously in women [31,33,34,35], NHP (rhesus) [36], pigtailed monkeys [37], and capuchin monkeys [32], and in mice [38]. In these species, the reserve of preantral follicles indicated the females reproductive life span.

Immunostaining for Caspase-3 was observed in oocytes, granulosa cells, and theca cells from the primordial to preovulatory follicles in both species studied. Similar results have been described in the ovaries of women, with the same ovarian cellular components [39,40,41]. Thus, after the initial formation of the antrum, the activation of Caspase-3 represents the physiological effect of the follicle during atresia [42]. In the present study, the highest frequency of immunostaining for Caspase-3 was in the granulosa cells of the preantral and antral follicles. This indicates the occurrence of early follicle apoptosis in the oocytes and granulosa cells of the antral follicles [43,44]. In contrast, there was no immunostaining indicative of apoptosis in the preantral and non-atretic antral follicles; a similar result has been observed in the ovaries of rats [39,45]. This suggests that these follicles tend to develop. These findings indicate that apoptosis is evidence of follicular depletion and atresia during follicular dynamics in the ovaries of this study. These findings have also been observed in the ovaries of women and baboons [46]. According to Tilly et al. [47], apoptosis and the altered expression of the related genes are associated with ovarian cell degeneration in all vertebrate species. The same pattern was observed in the present study.

The presence of BCL-2 in granulosa cells and oocytes in the three sexual phases tested in this study was relevant, as a similar result has been observed in the ovaries of women and rats [48]. However, when comparing BCL-2 immunostaining in Ma’s night monkey and large-headed capuchin ovaries, we observed a more significant value in the follicular phase, specifically in the antral follicles. Previous studies have found a greater expression of BCL-2 in the follicular phase of women [49,50] and suggest that this expression, in the menstrual cycle, is related to hormone regulation. Our results suggest that BCL-2 has a regulatory role in the cellular dynamics of the ovarian cycle in both species in this study.

The results of the quantified granulosa cells, ovarian follicles, and oocytes immunostained for BCL-2 demonstrated its protective role against the advancement of the apoptotic process. This aligns with previous studies that have shown that BCL-2 acts as an ovarian anti-apoptotic factor [48,51,52,53]. Therefore, in this study, we suggest that the apoptotic process was suppressed by the action of BCL-2 to allow follicular repair or development, especially in the follicular phase. In contrast, we observed intense immunostaining for PCNA from the beginning of the follicular growth period. Similar results were observed in rats [54] and pigs [55]. According to Hirshfield [51], an intense proliferative activity of granulosa cells occurs shortly before the formation of the antrum. Thus, due to the increase in antral follicles, immunostaining for PCNA is reflected in the high rates of granulosa cell proliferation [55].

In the present study, intense immunostaining for PCNA was observed in oocytes from large antral follicles. A similar result was reported by Downey et al. [56], who linked this to DNA damage repair. The highest number of PCNA-positive granulosa cells, oocytes, and antral follicles occurred mostly in the follicular phase. These data corroborate the data previously observed in rat [54] and baboon ovaries [21]. Therefore, the results of the present study confirm that PCNA is an early marker of the onset of follicular growth, which contributes to a better understanding of the proliferative mechanisms involved in folliculogenesis of the species studied.

In this study, immunostaining for AMH was observed in the granulosa cells of the preantral follicles, specifically the primary and antral follicles, particularly in the follicular phase. Such findings were also observed in women [57,58], mice [59,60,61], and NHP (bonnet macaque [23]; white-tufted-ear Marmoset [62]; squirrel monkey [63,64]; olive baboon [65]). AMH is produced by granulosa cells from follicles recruited to dominance and is likely involved in the regulation of follicular steroidogenesis [66]. Furthermore, AMH can act as an indirect serum marker of ovarian reserve and is considered a potential biomarker of ovarian follicular status [67,68,69,70]. Thus, we suggest that AMH acts as a regulator in the maturation of preantral and antral follicles during the ovarian cycle of the species in this study.

The absence of immunostaining for AMH was observed in the primordial follicles in the species studied. However, in the primary follicles and small antrals, the highest expression of AMH occurred close to the antrum and in the large antrals around the oocyte. This aligns with previous observations in other species such as mice [60,71] and rats [72,73,74]. It has been reported that follicular growth exerts inhibitory feedback on immature follicles, such as primordial follicles [22,75,76,77]. Thus, our results confirm that the influence of AMH on the ovary occurs from the primary follicle, favoring the recruitment of follicles capable of development.

## 5. Conclusions

The immunoreaction for cleaved Caspase-3, BCL-2, PCNA, and AMH indicates that the quantification of ovarian follicles, granulosa cells, and oocytes can assist in the evaluation of folliculogenesis in the ovaries of the species studied. These results aid our understanding of the action of proteins in folliculogenesis, which plays an important role in the gonadal remodeling process. This facilitates cell selection, inhibition, proliferation, repair, and elimination in follicular maturation. Therefore, our results improve the understanding of the events involved in ovarian dynamics and gamete production in these species.

## Figures and Tables

**Figure 1 animals-13-01051-f001:**
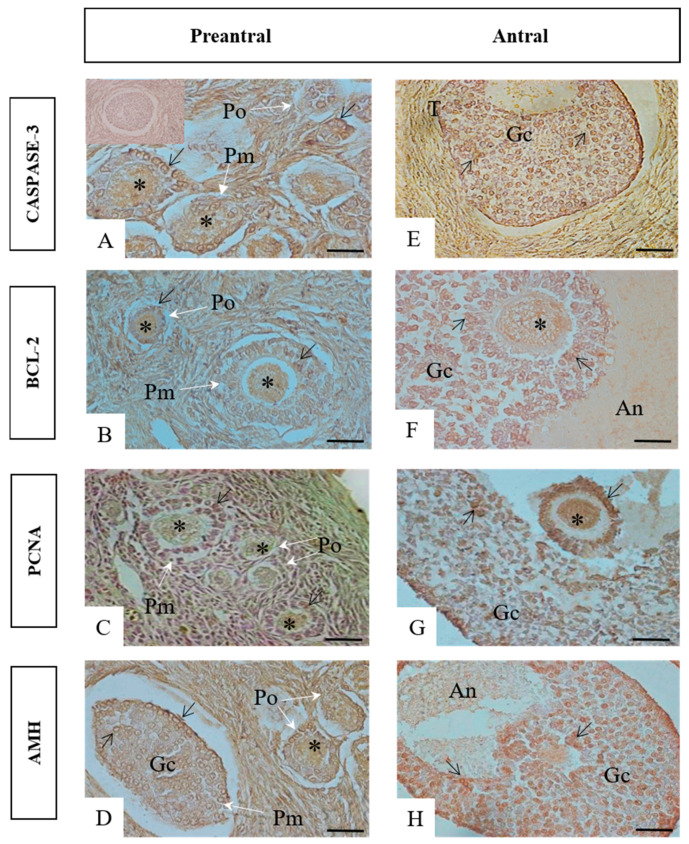
Photomicrographs of ovarian immunostaining for cleaved Caspase-3, BCL-2, PCNA, and AMH in granulosa cells and oocytes of the primordial and primary follicles (**A**–**D**) and small and large antral follicles (**E**–**H**) of Ma’s night monkeys (*Aotus nancymae*) and large-head capuchins (*Sapajus macrocephalus*). A: Negative control; Abbreviations: Po: Primordial follicle; Pm: Primary follicle; T: Theca layers; Gc: Granulosa cells; An: Antrum; *: oocytes. Arrow: immunostained granulosa cells. Hematoxylin–Eosin. Bar Scale: 25 µm.

**Figure 2 animals-13-01051-f002:**
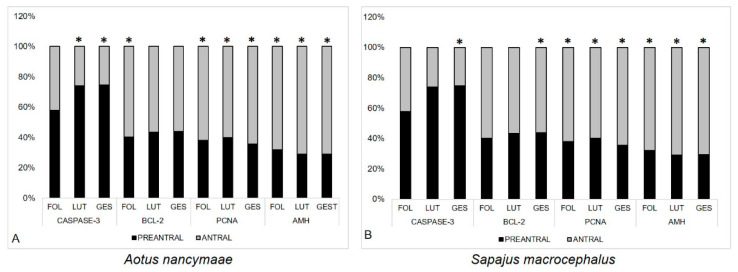
Percentage of follicles (preantral and antral) immunostained for Caspase-3, BCL-2, PCNA, and AMH, according to the sexual phases (FOL—follicular, LUT—luteal and GES—gestational). (**A**) Ma’s night monkeys (*Aotus nancymae*, *n* = 6) and (**B**) head-large capuchins (*Sapajus macrocephalus*, *n* = 6). (*) indicate that there is a statistically significant difference between preantral and antral follicles, according to the stage (Binomial Test. *p* < 0.01).

**Figure 3 animals-13-01051-f003:**
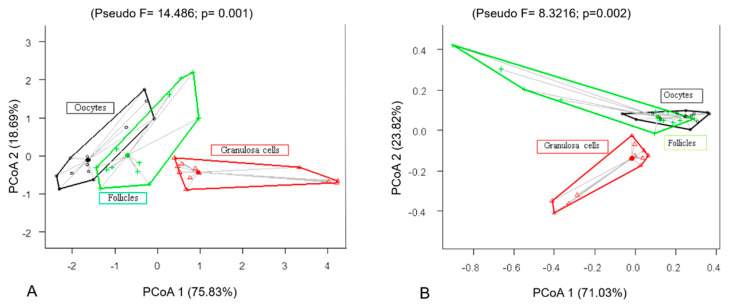
Principal coordinate analysis (PCoA) of the variation in the follicular cells, follicle, and oocytes in cleaved Caspase-3, BCL-2, PCNA, and AMH. (**A**) Ma’s night monkeys (*Aotus nancymae*, *n* = 6) and (**B**) head-large capuchins (*Sapajus macrocephalus*, *n* = 6).

**Figure 4 animals-13-01051-f004:**
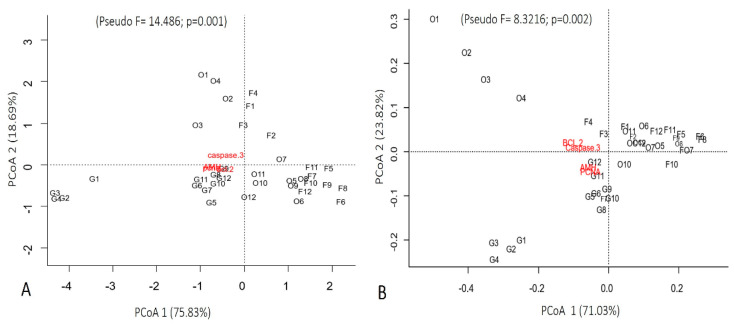
Immunohistochemical variables (Caspase-3, BCL-2, PCNA, and AMH), sorted by the principal coordinate analysis (PCoA) according to follicular types. (**A**) Ma’s night monkeys (*Aotus nancymae*, *n* = 6) and (**B**) head-large capuchins (*Sapajus macrocephalus*, *n* = 6). O: Oocytes; F: Follicles: G: Granulosa cells.

**Table 1 animals-13-01051-t001:** Criteria for ovarian follicle classification in Ma’s night monkeys (*Aotus nancymae*, *n* = 6) and large-head capuchins (*Sapajus macrocephalus*, *n* = 6) by Mayor et al. (2016, 2015b, 2019).

Species	Follicle Structures	Ovarian Follicles			
		Preantral (Absence of Antrum)		Antral (Presence of Antrum)	
		Primordial	Primary	Small	Large
*Aotus* *nancymae*	Diameter (µm)	35.26 ± 7.13	109.35 ± 25.03	275.36 ± 67.57	728.51 ± 126.75
	Layers	1	2–6	7–10	>10
*Sapajus* *macrocephalus*	Diameter (µm)	33.76 ± 2.27	112.30 ± 31.43	238.84 ± 67.48	858.54 ± 149.05
	Layers	1	2–6	7–10	>10

**Table 2 animals-13-01051-t002:** Quantification of the ovaries of Ma’s night monkeys (*Aotus nancymae*, *n* = 6) and large-head capuchins (*Sapajus macrocephalus*, *n* = 6) according to the sexual phase.

Species	Sexual Phases	Ovarian Follicles			
		Pre-Antral		Antral	
		Primordial	Primary	Small	Large
*Aotus* *nancymae*	Follicular	147.4 ± 6.6 ^a^	83.9 ± 6.8 ^ab^	17.6 ± 1.0 ^bc^	5.0 ± 0.1 ^c^
	Luteal	107.5 ±15.7 ^a^	51.1 ± 1.6 ^ab^	14.0 ± 0.5 ^bc^	3.8 ± 0.1 ^c^
	Gestational	123.2 ± 6.6 ^a^	49.7 ± 6.2 ^ab^	13.7 ± 0.1 ^bc^	3.7 ± 0.1 ^c^
*Sapajus* *macrocephalus*	Follicular	141.0 ± 11.2 ^a^	70.8 ± 1.6 ^ab^	16.6 ± 0.7 ^bc^	5.4 ± 0.1 ^c^
	Luteal	104.5 ±6.5 ^a^	45.8 ± 2.3 ^ab^	13.0 ± 0.5 ^bc^	3.6 ± 0.07 ^c^
	Gestational	135.1 ± 8.7 ^a^	53.4 ± 1.8 ^ab^	13.4 ± 0.4 ^bc^	3.4± 0.06 ^c^

^abc^ Different superscript letters on the same line indicate statistical difference.

**Table 3 animals-13-01051-t003:** Number of ovarian follicles (Ov. follicles), granulosa cells, and oocytes immunostained for Caspase-3, BCL-2, PCNA, and AMH by sexual phases in Ma’s night monkeys (*Aotus nancymae*, *n* = 6) and large-head capuchins (*Sapajus macrocephalus*, *n* = 6).

Species	Immunoreaction	Quantification	Sexual Phases		
			Follicular	Luteal	Gestational
*Aotus* *nancymae*	Caspase-3	Ov. follicles	0.64 ± 0.07 ^a^	0.27 ± 0.07 ^b^	0.30 ± 0.04 ^ab^
		Granulosa cells	0.61 ± 0.02 ^a^	0.44 ± 0.04 ^b^	0.46 ± 0.02 ^ab^
		Oocytes	0.79 ± 0.09 ^a^	0.33 ± 0.08 ^b^	0.37 ± 0.03 ^ab^
	BCL-2	Ov. follicles	0.23 ± 0.04	0.19 ± 0.03	0.22 ± 0.04
		Granulosa cells	0.51 ± 0.04 ^a^	0.36 ± 0.03 ^ab^	0.35 ± 0.03 ^b^
		Oocytes	0.28 ± 0.04	0.23 ± 0.02	0.30 ± 0.07
	PCNA	Ov. follicles	0.24 ± 0.02 ^a^	0.18 ± 0.01 ^b^	0.19 ± 0.01 ^ab^
		Granulosa cells	0.65 ± 0.04 ^a^	0.42 ± 0.01 ^ab^	0.36 ± 0.03 ^b^
		Oocytes	0.32 ± 0.03 ^a^	0.24 ± 0.03 ^b^	0.26 ± 0.01 ^ab^
	AMH	Ov. follicles	0.28 ± 0.03 ^a^	0.16 ± 0.02 ^b^	0.19 ± 0.03 ^ab^
		Granulosa cells	0.60 ± 0.07 ^a^	0.32 ± 0.01 ^b^	0.33 ± 0.02 ^ab^
		Oocytes	0.32 ± 0.04 ^a^	0.22 ± 0.03 ^b^	0.25 ± 0.05 ^ab^
*Sapajus* *macrocephalus*	Caspase-3	Ov. follicles	0.52 ± 0.13 ^a^	0.24 ± 0.04 ^b^	0.26 ± 0.05 ^ab^
		Granulosa cells	0.55 ± 0.03 ^a^	0.39 ± 0.02 ^b^	0.37 ± 0.04 ^ab^
		Oocytes	0.77 ± 0.09 ^a^	0.30 ± 0.03 ^b^	0.30 ± 0.02 ^ab^
	BCL-2	Ov. follicles	0.30 ± 0.02 ^a^	0.17 ± 0.01 ^b^	0.2 ±0.02 ^ab^
		Granulosa cells	0.55 ± 0.04 ^a^	0.34 ± 0.02 ^b^	0.36 ± 0.03 ^ab^
		Oocytes	0.97 ± 0.24 ^a^	0.26 ± 0.04 ^b^	0.36 ± 0.01 ^ab^
	PCNA	Ov. follicles	0.24 ± 0.01	0.20 ± 0.01	0.23 ± 0.01
		Granulosa cells	0.69 ± 0.03 ^a^	0.48 ± 0.02 ^ab^	0.38 ± 0.02 ^b^
		Oocytes	0.28 ± 0.01 ^ab^	0.24 ± 0.01 ^a^	0.30 ± 0.03 ^b^
	AMH	Ov. follicles	0.27 ± 0.02 ^a^	0.16 ± 0.01 ^b^	0.18 ± 0.03 ^ab^
		Granulosa cells	0.59 ± 0.04 ^a^	0.31 ± 0.01 ^b^	0.34 ± 0.03 ^ab^
		Oocytes	0.32 ± 0.03 ^a^	0.19 ± 0.02 ^b^	0.21 ± 0.03 ^ab^

^abc^ Different superscript letters on the same line indicate statistical difference.

## Data Availability

Not applicable.

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
