# Peer review of "Follicular Atresia, Cell Proliferation, and Anti-Mullerian Hormone in Two Neotropical Primates (Aotus nancymae and Sapajus macrocephalus)"

_animals, 2023, doi:10.3390/ani13061051_

Round 1
Reviewer 1 Report
Materials and Methods
Please replace “2.1. Study sites and ethical procedures” (line 70) to “2.1. Study sites and animals”, ethical procedures were already indicated on “Institutional Review Board Statement” (lines 331-337).
I suggest the common name Ma's Night Monkey for A. nancymae.
If the "large-headed capuchins ovaries were collected as waste organs from subsistence hunters in the Yavarí-Mirín River" (lines 74-76). Thus, it is not possible to know about the female ages, right? However, in the methodology, it must be important to indicate that ovaries were collected in post-pubertal age females, by the ovaries features used. Regarding captive females (owl monkeys), please provide more information about captive animals, age of females for instance. Is it possible to include this information?
Please use italic on scientific names Table 1 Lines (94-95).
Results
Please use italic on scientific names Table 2 Lines (154-155), Table 3(185-186), Figure 2 (line 204), Figure 3 (216-217), and Figure 4 (lines 233-234). Please review all documents.
Table 2, replace “Follicular” to “Follicular”. Replace Aotus Nancymae to nancymae.
Table 3, replace “Granulos cells” to “Granulosa cells”
Discussion
The authors started the discussion talking that: "A greater number of preantral follicles (primordial and primary) was observed in relation to antrals in the two species studied". In sequence, they said: "Therefore, we believe that the quantity of preantral follicles present is directly related to ovarian reserve maintenance in both species in this study. This has been previously reported in women [31,33-35], NHP (rhesus, [36]; pigtailed monkeys, [37], and capuchin monkeys, [32], and in mice [38]." However, which is this relationship in post puberal female?
Usually, New World primate ovaries are proportionally larger than in Old World NHPs because of the large amount of interstitial glandular tissue. For instance, the extensive mass of luteinic tissue can be responsible for the high steroid levels detected in the owl monkey's plasma and urine (progesterone, for example). Please clarify which is the relationship of that information and the findings in the present study. In this context, how is it possible to establish a relationship between the quantification of ovarian follicles, granulosa cells and oocytes with the sexual phases (follicular, luteal and Gestational).
Author Response
Materials and Methods Please replace “2.1. Study sites and ethical procedures” (line 70) to “2.1. Study sites and animals”, ethical procedures were already indicated on “Institutional Review Board Statement” (lines 331-337). I suggest the common name Ma's Night Monkey for A. nancymae. If the "large-headed capuchins ovaries were collected as waste organs from subsistence hunters in the Yavarí-Mirín River" (lines 74-76). Thus, it is not possible to know about the female ages, right? However, in the methodology, it must be important to indicate that ovaries were collected in post-pubertal age females, by the ovaries features used. Regarding captive females (owl monkeys), please provide more information about captive animals, age of females for instance. Is it possible to include this information? Please use italic on scientific names Table 1 Lines (94-95). Results Please use italic on scientific names Table 2 Lines (154-155), Table 3(185-186), Figure 2 (line 204), Figure 3 (216-217), and Figure 4 (lines 233-234). Please review all documents. Table 2, replace “Follicular” to “Follicular”. Replace Aotus Nancymae to nancymae. Table 3, replace “Granulos cells” to “Granulosa cells” Response: All changes were mark in yellow on the text Discussion The authors started the discussion talking that: "A greater number of preantral follicles (primordial and primary) was observed in relation to antrals in the two species studied". In sequence, they said: "Therefore, we believe that the quantity of preantral follicles present is directly related to ovarian reserve maintenance in both species in this study. This has been previously reported in women [31,33-35], NHP (rhesus, [36]; pigtailed monkeys, [37], and capuchin monkeys, [32], and in mice [38]." However, which is this relationship in post puberal female? We complete the sentence, for better understanding, as follows, please see lines 247-248 Usually, New World primate ovaries are proportionally larger than in Old World NHPs because of the large amount of interstitial glandular tissue. For instance, the extensive mass of luteinic tissue can be responsible for the high steroid levels detected in the owl monkey's plasma and urine (progesterone, for example). Please clarify which is the relationship of that information and the findings in the present study. In this context, how is it possible to establish a relationship between the quantification of ovarian follicles, granulosa cells and oocytes with the sexual phases (follicular, luteal and Gestational). In this manuscript, no observations were made in the luteal tissue, which does not allow any comparison, since the main objective was the study of follicular atresia. This process occurs during all reproductive stages. This dynamic process occurs in follicular waves, related to certain factors and hormones, which allows observing follicular activity during each reproductive phase. Consequently, the same is verified for oocytes and granulosa cells.
Reviewer 2 Report
General comments:
Studies on follicular atresia are not new. However, to authors credit, they have specifically focused on the expression of certain markers (Caspase-3, BCL-2, PCNA, and AMH proteins) in the process of follicular atresia in two primate species (owl monkey and large-headed capuchin). Particularly, using information from a mouse model for PCNA (reference 20) is somewhat new. Ovaries (6 of each) were gathered from animals kept in captivity and from hunters, respectively. Expression of markers were studied in oocytes, theca, and granulosa cells in preantral and antral follicles. As expected, immunostaining was detected in preantral follicles and antral follicles. Expression was tissue specific and reproductive phase specific. Studies on primates are limited due to availability of animals. From this perspective, it makes the study relevant. Authors have cited adequate and pertinent references. Methodology (immunostaining and data analyses) appears acceptable and presentation (tables and figures) is appropriate.
Specific comment:
Authors claim that findings are contributing to greater understanding of the morphofunctional processes that act in follicular selection, recruitment, and degeneration of these species. Problem with this statement is neither they are contributing to greater understanding nor to all these processes. Authors aim was to study ‘follicular atresia’ and hence they need to very careful in wording their conclusion.
line 18 & all through the manuscript: sexual to reproductive
line 22: species in italics
lines 33 & 76: nonpregnant, be consistent
lines 43 & 79: menstrual cycle, be consistent
lines 54: delete hyphen
line 121 & all through the manuscript: P no italics, also space after P be consistent
line 160: A: is it an insert? Delete insert
lines 165 & 169: move P values next to differences or higher and delete significant
line 196: not sure about the use of Pseudo here & all through the manuscript, readers may need help to decipher, needs some explanation
line 240: delete hyphen
Author Response
Comments and Suggestions for Authors (REVIEWER 2)
General comments:
Studies on follicular atresia are not new. However, to authors credit, they have specifically focused on the expression of certain markers (Caspase-3, BCL-2, PCNA, and AMH proteins) in the process of follicular atresia in two primate species (owl monkey and large-headed capuchin). Particularly, using information from a mouse model for PCNA (reference 20) is somewhat new. Ovaries (6 of each) were gathered from animals kept in captivity and from hunters, respectively. Expression of markers were studied in oocytes, theca, and granulosa cells in preantral and antral follicles. As expected, immunostaining was detected in preantral follicles and antral follicles. Expression was tissue specific and reproductive phase specific. Studies on primates are limited due to availability of animals. From this perspective, it makes the study relevant. Authors have cited adequate and pertinent references. Methodology (immunostaining and data analyses) appears acceptable and presentation (tables and figures) is appropriate.
Specific comment:
Authors claim that findings are contributing to greater understanding of the morphofunctional processes that act in follicular selection, recruitment, and degeneration of these species. Problem with this statement is neither they are contributing to greater understanding nor to all these processes. Authors aim was to study ‘follicular atresia’ and hence they need to very careful in wording their conclusion.
Abstract: We have restructured this phase for the following:
Our results contribute to understand the ovarian follicular selection, recruitment, and degeneration of these species.
line 18 & all through the manuscript: sexual to reproductive
line 22: species in italics
lines 33 & 76: nonpregnant, be consistent
lines 43 & 79: menstrual cycle, be consistent
lines 54: delete hyphen
line 121 & all through the manuscript: P no italics, also space after P be consistent
line 160: A: is it an insert? Delete insert
lines 165 & 169: move P values next to differences or higher and delete significant
line 196: not sure about the use of Pseudo here & all through the manuscript, readers may need help to decipher, needs some explanation - Pseudo F is an Index statistic. Please, could you read about it here: https://stats.stackexchange.com/questions/448188/what-is-the-difference-between-the-pseudo-f-index-and-the-f-statistic Pseudo F
line 240: delete hyphen
Response: All changes were mark in blue on the text
